# Transforming IoT Events to Meaningful Business Events on the Edge: Implementation for Smart Farming Application

**Dimitris Gkoulis *** , **Cleopatra Bardaki** , **George Kousiouris** and **Mara Nikolaidou**

Department of Informatics and Telematics, Harokopio University of Athens, 17778 Athens, Greece
* Correspondence: gkoulis@hua.gr

**Abstract:** This paper focuses on Internet of Things (IoT) architectures and knowledge generation out of streams of events as the primary elements concerning the creation of user-centric IoT services. We provide a general, symmetrical IoT architecture, which enables two-way bidirectional communication between things and users within an application domain. We focus on two main components of the architecture (i.e., Event Engine and Process Engine) that handle event transformation by implementing parametric Complex Event Processing (CEP). More specifically, we describe and implement the transformation cycle of events starting from raw IoT data to their processing and transformation of events for calculating information that we need in an IoT-enabled application context. The implementation includes a library of composite transformations grouping the gradual and sequential steps for transforming basic IoT events into business events, which include ingestion, event splitting, and calculation of measurements' average value. The appropriateness and possibility of inclusion and integration of the implementation in an IoT environment are demonstrated by providing our implementation for a smart farming application domain with four scenarios that each reflect a user's requirements. Further, we discuss the quality properties of each scenario. Ultimately, we propose an IoT architecture and, specifically, a parametric CEP model and implementation for future researchers and practitioners who aspire to build IoT applications.

**Keywords:** Internet of Things (IoT); Complex Event Processing; event-driven architecture; smart farming

## 1. Introduction

The Internet of Things (IoT) is a novel paradigm that enables interconnection among devices anytime, anywhere on the planet, thus providing the Internet's advantages in all aspects of human activity [1]. Moreover, IoT has fundamentally shifted how we interact with our surroundings; the ability to monitor and manage things in the physical world through the digital world creates not only new possibilities and opportunities but also challenges [1,2].

From a technical point of view, the most significant issues [2] that emerge from the IoT literature concern: (i) massive scaling, (ii) architectures and dependencies creating complex IoT systems, (iii) knowledge creation out of data, (iv) robustness as a broad concept that includes the Quality of Service, (v) openness, (vi) interoperability, (vii) security, (viii) privacy, and, last but not least, (ix) the notion of human-in-the-loop.

All challenges are equally important; however, in this paper, we focus on IoT architectures, knowledge generation out of data, and Quality of Service as the primary elements concerning creating user-centric IoT systems.

More specifically, there are several approaches to what constitutes an IoT architecture, its structural elements, and the characteristics it should have. Analyzing these approaches, we conclude that more effort must be given to how the data originating from the natural environment is transformed into information and, ultimately, into knowledge beneficial for humans and their interaction with the natural world. Thus, we provide a general,

symmetrical architecture, which enables two-way bidirectional communication between things and users within an application domain; and focus on two main components of the architecture (i.e., Event Engine and Process Engine) that handle event transformation and implement parametric Complex Event Processing (CEP). More specifically, we describe and implement the transformation cycle of events starting from raw IoT data to their processing and transformation of events for calculating a quantity that we need in an IoT-enabled application context. We showcase our implementation considering an IoT-enabled greenhouse in a smart farming application domain.

In Section 2, we summarize the related work. In Section 3, we detail the proposed architecture, and in Section 4, we present the Complex Event Processing model that enhances the proposed architecture. In Section 5, we present an example of implementation of the architecture based on the introduced Complex Event Processing Model to manage an Internet of Things environment. Next, in Section 6, we present a realistic example of the implementation in a smart farming domain to exemplify the suitability of the architecture and the Complex Event Processing Model. Finally, we discuss the quality properties of the implementation and conclude the paper.

## 2. Related Work

Research has attempted to define and delineate the Internet of Things [3–8] as well as its structural elements. In the area of this study, which is the creation and operation of IoT services for managing devices, there are several opinions and research on IoT architectures [9–12]. Specifically, many studies [13–15] suggest the exploitation of Microservices Architecture [16–18], Service-Oriented Architecture [19], and, more generally, Event-Driven Architecture [20–22] for the creation of IoT services.

Event-Driven Architecture (EDA) refers to an abstract concept of a responsive system that promotes the production, detection, consumption of, and reaction to events. Consequently, research [13–15,23,24] suggests the utilization of EDA for creating IoT systems. An event-driven system typically consists of event producers, event consumers, and event channels. Event producers and consumers may be services and IoT devices such as sensors and actuators. Event producers generate streams of events that are distributed through event channels to multiple event consumers. Such a system uses the publish–subscribe pattern [25] for communication and control and provides concurrent responsive processing. Consequently, it is especially suitable for applications that use loosely coupled real-time communication and support sensing. Thereby, users and systems can respond to and process these events in a rapid and appropriate manner. Event processing usually includes Event Stream Processing (ESP) [26,27] and Complex Event Processing (CEP) [27–29].

A big issue for many studies in the domain of IoT [30–36], including more general studies as those mentioned before, is management of the massive influx of data generated by devices in the physical environment while sensing and actuating [37]. Those data are treated as events, as they usually define a significant change in state and must be appropriately processed in a timely and responsive manner. In order to be meaningful in the application context, the events must be transformed. In its broadest definition, a transformation [38] refers to any operation that takes as input a single event or stream of events and produces a single event or stream of events as its output. The produced type of event or stream of events usually has different semantics than the input event or stream of events.

A set of techniques and tools that provides visibility and control over event-driven architectures is Complex Event Processing [29]. According to G. Cugola and A. Margara [27], Complex Event Processing (CEP) [29] belongs to the broader category of applications known as Information Flow Processing (IFP). Together with Data Stream Processing (DSP), they are the two most well-known and widespread application models, capable of timely processing of large amounts of information as it flows from the peripheral (e.g., the physical world) to the Center of the System. The Complex Event Processing model views flowing raw data as messages of events happening in the external world, which have to be pro-

cessed and transformed into higher-level events to understand what is happening in terms of domain knowledge. Regarding communication, CEP is primarily based on an interaction style known as publish–subscribe, a message-oriented interaction paradigm based on an indirect addressing style that comes in two types: topic and content-based. Finally, another defining aspect of CEP is that it emphasizes giving precise semantics to the information chunks being processed.

Some of the aforementioned studies support the potential of Complex Event Processing to extract high-level information from these data and thus enable users and systems to have effective, precise, and timely interactions with the physical environment [31,39]. However, although CEP is a mature and proven technology, its integration into the IoT world is either developing or facing several challenges [40,41], such as providing precise semantics to streams of events in a scalable way, performing optimally CEP on the edge, establishing a universal expressiveness language, transforming data to meaningful information in the application domain, and more. In this context, we propose (i) an architecture for establishing bidirectional communication between IoT Physical and the users, (ii) a CEP model to enhance the architecture by enabling the flexible management of an IoT environment through transforming data into meaningful information, and (iii) implementation of the architecture and CEP model with the aim of operationalizing the procedures for handling common situations. Moreover, we present a simple but realistic case study to exemplify the implementation and showcase our contribution.

### 3. An Event-Based IoT Architecture

The proposed architecture draws on and extends the architecture presented in [42]. It is technology-agnostic and, at the same time, infrastructure-agnostic, serving full or hybrid integration in any computing model (edge, fog, cloud). Further, it is a symmetrical representation of the original architecture [42], and while it retains many of its dimensions, layers, and components, the assumptions and limitations regarding how services are created in the corresponding layer have been removed. The architecture serves any IoT environment, including IoT devices, which are divided into two main categories: Sensors and Actuators. Therefore, two basic notions are served: Capture and Act.

The architecture is event-driven, and the communication of the components is asynchronous. The essential notion of the architecture is the event [43]. An event is a notable thing that happens inside or outside the defined business. It may signify a measurement, a situation, a state, a trigger, a signal, a threshold, a problem, an opportunity, and more. In the context of IoT, those notable things usually serve the notions of sensing and actuating. As a term, event is often used interchangeably to refer to both the specification (definition) of the event and each individual occurrence (instance) of the event. Furthermore, for an event to be meaningful to downstream subscribers (human, hardware, and software), it is imperative that the event (and its attributes as a whole) is specified in business terms, not data or application terms.

In the context of the proposed architecture, an event happens inside or outside the defined business and is immediately disseminated to all interested parties (human, hardware, and software). The interested parties evaluate the event and optionally take action. The event-driven action may include invocating a service, triggering a business process, or further information publication/syndication.

Next, we present the layers of the architecture with their corresponding components, as shown in Figure 1.

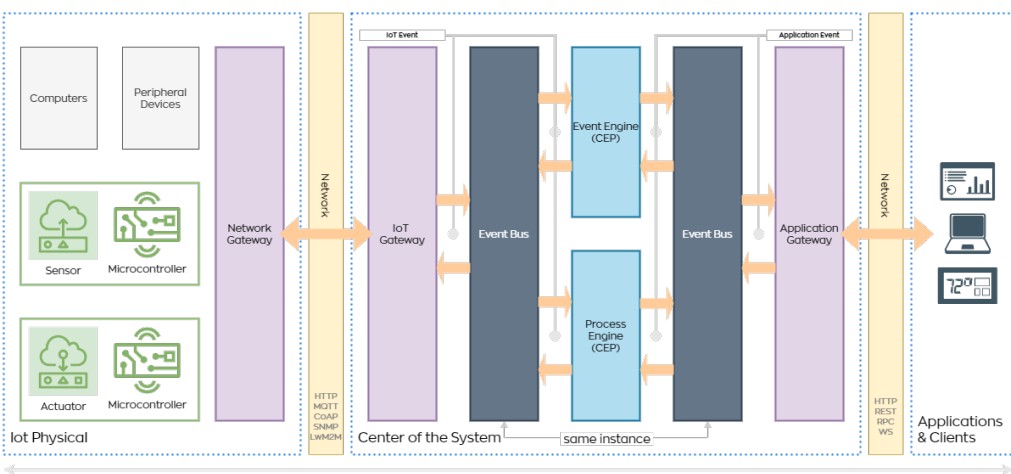

**Figure 1.** Architecture overview.

**IoT Physical** [42]: IoT Physical is a layer of the architecture and includes the physical world, IoT devices (Sensors and Actuators), Network Gateways and Routers, as well as peripheral computers on which programs are executed, usually composed microcontrollers. Representative examples: (i) An IoT device with sensing capabilities (sensor) that measures the air temperature and transmits it to the microcontroller every second, and the microcontroller transforms it and pushes it to a topic in an MQTT. (ii) An IoT device with actuating capabilities (actuator) that receives a binary command from the microcontroller to open or close the roof vent in a greenhouse. The microcontroller receives data in real time by subscribing to an MQTT topic.

**Applications and Clients** [42]: Application is a layer of the architecture and includes, among other things, mobile applications, front-end applications, clients, and interfaces with third-party systems. From this layer, the user can learn and manage the situation in the physical world. Representative example: A mobile application through which the user can see a greenhouse's temperature and remotely open or close the roof vent.

**Center of the System** [42]: The Center of the System is the layer that includes all those components that allow the management of the physical world as well as the creation of applications. It establishes communication with the IoT Physical and Applications and Clients layers, provides structures for creating event-driven services, supports interoperability, and ensures the correct flow of data in the direction for which it is intended. The Center of the System is essential for our research and consists of the following components.

**Event Bus** [42]: It is quite different from typical brokers and is shaped by a different set of forces. It is distributed at its core, fault-tolerant, focused on scale, throughput, and retention, and fundamentally consists of (i) message queuing, (ii) message processing, and, last but not least, (iii) message storage. All architecture components are built on a backbone of events, where events become both a trigger and a pattern for distributing states.

**Event Engine and Process Engine** [42]: The purpose of the components is to provide functionality for providing information and performing actions. In any case, raw data are converted into events, and events into meaningful events, on the backbone of which we create services. Event Engine transforms data from IoT Physical to IoT Events. IoT Events can be used by Process Engine to create Application Events with the final destination, the Applications and Clients layer. Process Engine transforms data from the Applications and Clients layer to Application Events. Event Engine can use Application Events to create IoT Events with the final destination, the IoT Physical layer. Both engines are distributed and implement Complex Event Processing to achieve their intended use.

**IoT Gateway** [42]: Connects the Center of the System layer with the IoT Physical layer, i.e., establishes direct communication with IoT Physical. The communication is standardized, based on a widely used and established protocol, and can be synchronous or asynchronous. It consists mainly of transports, which are services that establish two-

way communication with the IoT devices of IoT Physical, i.e., the physical environment. Transports may include MQTT, CoAP, LwM2M, and, last but not least, HTTP.

**Application Gateway** [42]: It acts as a single entry point for all applications and clients and provides an effective way to route to APIs through the Center of the System layer. Multiple Application Gateways may cater to the needs of different kinds of clients. For example, an Application Gateway can be a basic reserve proxy or even a complete API Management System that may provide cross-cutting concerns to APIs, such as security, monitoring, resiliency, smart and/or conditional routing. The architectural components are dynamic and heterogeneous. The number of their instances and physical or virtual locations changes dynamically, as well as their intercommunication protocols. Hence, the role of an Application Gateway is vital ,as it provides clients with a unified, consistent, and standardized way of accessing the Center of the System layer through the provided APIs.

By its nature, the proposed event-driven architecture is extremely loosely coupled and highly distributed. Although it is intended that the architecture be technology-agnostic, we propose utilizing Apache Kafka [44], an open source distributed event streaming protocol used for high-performance data pipelines, streaming analytics, data integration, and mission-critical applications. It follows the concept of dump broker/smart consumer, which puts all the focus on doing one thing in a highly effective manner, i.e., scalable and fault-tolerant data distribution. This concept shifts the implementation responsibility and the emerging logical complexity to the consumer side, which is very convenient for our approach, since we have all the offerings by the event broker focusing on data distribution and simultaneously have total control of the application focusing on the processing of data.

Overall, the proposed symmetrical architecture contributes significantly to the aim of this research, which is to describe and implement the transformation cycle of raw data to meaningful events that serve the requirements of the application domain. More specifically, data from IoT devices are progressively transformed into IoT Events and, finally, business events. Respective data from Applications and Clients are progressively transformed into Application Events and finally business events. Both types of events can trigger further transformations. As afore-described, two main components of the architecture (i.e., Event Engine and Process Engine) handle the event transformation and implement Complex Event Processing, which we describe next, and we then demonstrate implementation inspired by a smart farming application domain.

## 4. CEP Model: Transforming IoT Events to Meaningful Business Domain Events

Having presented an architecture for managing an IoT environment, we introduce Complex Event Processing for parametric progressive transformations that generate events that are meaningful in the context of an application domain. The proposed model is about a core concept, not a structure or a system. In other words, it is a conceptual framework that lays the foundations for creating and executing runtime business logic aware elements in the Center of the System layer of the proposed symmetrical functional architecture at the backbone of events and consequently establish asynchronous bi-directional communication between IoT and Applications based on a consistent, unified expression language for declaring Data and Processing Models in the domain context. In essence, the CEP model provides the ability to synthesize meaningful events that carry meaningful business information out of raw data and domain knowledge. The visual representation of the architecture and the CEP model is shown in Figure 2.

In Section 3, we presented an approach to how meaningfulness is elicited in the application domain context. However, meaningfulness in the context of CEP can be many things, including semantics, schema, correlations, associations, topics, origin, domain knowledge, data quality, and every aspect that distinguishes raw and/or transient data from meaningful information. These concepts collectively assemble a conceptually meaningful event. We clarify that we do not make any assumptions regarding the complexity of a meaningful event. Moreover, we do not introduce any limitations on what can be considered a mean-

ingful event. Conceptually, it is a transformation product that is more than a data transfer object (DTO) and has specific semantics in the context of the IoT.

**Figure 2.** Symmetrical layered Architecture enables parametric Complex Event Processing.

Events, topics, and events streams are the basic notions of a traditional topic-based publish–subscribe model. Applications, topologies, and processors, which are presented in the following paragraphs, are essential building blocks of Complex Event Processing. All presented terms collectively constitute an expressive visual language that describes how information has to be processed and finally transformed into something meaningful. Building blocks can be parametric to boot. The developer can define which parts of the above elements are parameterizable, parameterize them, and get different instantiations of the same application, topology, or processor definition. The parameterization is conducted at application or topology level but may affect all encapsulated elements that make up the respective application or topology.

The concept of parametrization is broad. Without introducing any limitations on what constitutes parameterization and parametric elements, we present some examples of parameters: the number of devices to be taken into account in the calculation of a value, a statistical method (standard deviation or mean, minimum or maximum), the duration of a time window, how to handle an error, how often to retrieve a calculated value (given that many values are produced), and more.

An **event** represents something that happened in the observed world or, more generally, any kind of data that reflects some knowledge generated by the origin. Technically, it is a record or, more precisely, a message distributed across the architectural runtime

elements. In its most basic conceptual form, an event has a key, value, timestamp, and optional metadata.

Events are organized and live in **topics** that are named resources acting as virtual directories of records. Events are written to a topic by producers, which are client applications that publish events and read from a topic by consumers that are also client applications consuming and processing events.

Events, which are the cornerstone of the proposed architecture, are processed in processing pipelines consisting of multiple stages. Raw input data are consumed from the topics and then processed (aggregated, enriched) or otherwise transformed and forwarded into new topics for further consumption or follow-up processing. Such processing pipelines create (conceptual) directed acyclic graphs (DAGs) of real-time data flows based on individual topics.

An **event stream** (henceforth stream), in turn, is an essential abstraction of CEP: it represents an unbounded, continuously updating data set representing a flow with chunks of information. Events that are part of the same stream are neither necessarily ordered nor of the same kind. Instead, they are information chunks with semantics and relationships (including total or partial ordering, in some cases). Technically, an event stream is an ordered, replayable, and fault-tolerant sequence of immutable data records, where a data record is defined as a key–value pair.

An **event stream processing application** (henceforth application) is a dynamically defined computational logic through one or more event stream processor topologies, where an **event stream processor topology** (henceforth topology) is a graph (DAG) of event stream processors (graph nodes) that are connected by event streams (graph edges).

An **event stream processor** (henceforth processor) is a node in the topology. It represents a processing step to transform data in streams by receiving one input record at a time from its upstream processors in the topology, applying its operation to this, and may subsequently produce one or more output records for its downstream processors.

There are two special kinds of processors:

A **source processor** does not have any upstream processors. It produces an input stream to its topology from one or multiple topics by consuming records from these topics and forwarding them to its downstream processors.

A **sink processor** does not have downstream processors. It sends any received records from its upstream processors to a specified topic.

Each processor reflects a transformation, i.e., processing.

Some of the most well-known transformations are *map*, *filter*, *group*, *join*, and others, such as those presented in the Apache Kafka Streams Developer Guide [44]. The above transformations are primitive and are usually encountered as they are with expected behavior, input, and output. However, a transformation can be complex. It can have a state and communicate with a database or external system. For example, a complex transformation is a time-windowed average value calculation considering filtered values based on some condition. Another example of a complex transformation is detecting an anomaly in a data flow or performed by some other component, e.g., an external machine learning system.

An application may have one or more topologies. A topology may have one or more sub-topologies. A sub-topology may have one or more processors. Collectively, they constitute a set of processing rules describing how incoming streams of events must be processed to produce new streams as outputs in a timely manner.

Technically, there are two ways to define the topology: (i) the DSL, which provides the most common transformation operations such as map, filter, join, and aggregation and (ii) the low-level processor contract, which allows developers to define and implement custom processor as well as to interact with state stores.

Other technical details, such as data interchange, serialization, deserialization, batch processing, topics, processing guarantees, and others, are then presented in Section 5,

which follows, while an exemplification of the overall architecture is presented through some scenarios.

## 5. Proposed Implementation for Parametric Event Transformation

Here, we present the implementation of the architecture application that serves an IoT-enabled application context. We assume that the application context is a smart farming case with an IoT-enabled greenhouse, and we provide the implementation that serves this case and can be easily generalized to any IoT-enabled context. Our approach is comprehensive, but we place particular emphasis on how the data are parametrically transformed following a bottom-up approach. More specifically, this implementation will show the transformation cycle of events starting from raw IoT data to their processing and transformation of events for calculating a quantity that we need in the examined IoT-enabled application context.

First, we present some assumptions that are necessary for proper conception and assessment of the physical environment. We also present a fundamental example of implementation of specific layers and components of the proposed architecture in Section 3, which include the IoT Physical layer, the IoT Gateway, Event Bus, which is a Kafka broker, and, most notably, the Event Engine.

In the physical context, a peripheral structure is installed that allows the connection of all computers and IoT devices to the Internet. An installed and configured local network allows secure communication (both wired and wireless) between IoT devices and computers. There are potent sensors with excellent features on the market. However, the implementation uses a simple, basic custom implementation for each IoT device (i.e., a sensor) in the physical context that can serve various IoT-enabled applications in different application contexts. In our physical context, our setup includes a standard, highly used DHT20 sensor for collecting air temperature and humidity and a Raspberry Pi single-board computer for deploying and running microcontrollers. The DHT20 sensor continuously measures the temperature and allows us to read the measured values as often as we like. It has an overall accuracy of ±3% relative humidity and ±0.5 °C. It also uses standard I2C, so using it with any Arduino or Linux/Raspberry Pi board is straightforward. The DHT20 sensor is connected to a Raspberry Pi, which offers computational capabilities and a complete and self-contained Wi-Fi networking solution.

In Raspberry Pi, a Python application based on CircuitPython (a library designed to program microcontrollers easily) runs and acts as the microcontroller. The microcontroller collects data from the DHT20 sensor in a specified interval, performs fundamental data transformations, and finally pushes the transformed data with the measurement values and the necessary metadata to the IoT Gateway with a POST HTTP request. The microcontroller includes useful information such as the measurement point, the measurement zone, i.e., the location of the sensor, the measurement time, and more, thus providing useful data to the components that will use the measurement data, notably the IoT Gateway and the Event Engine. These metadata allow the Event Engine to perform complex tasks such as grouping, time windowing, and more, as described in Section 4. Each IoT device has a finite operating autonomy and charging capability. Both autonomy and charging are outside the scope of this research. The above description applies to each IoT device, i.e., the sensor, located in the greenhouse.

IoT Gateway specifications include support for multiple types of transport, such as HTTP, MQTT, CoAP, SNMP, LwM2M, and more. However, the current implementation is simple, and there is no requirement for two-way communication of the IoT device with the application. Therefore, we use HTTP Transport to forward the IoT device measurements to the IoT Gateway via HTTP. In HTTP Transport of the IoT Gateway, there is a REST Controller. The REST Controller exposes one endpoint that accepts only JSON POST requests. Moreover, the REST Controller can forward and receive messages from Kafka through the respective producers and consumers implemented in the IoT Gateway. The relative path of the REST Controller endpoint, i.e., the URI, is "sensor–telemetry–events". This exposed resource receives the JSON POST request and validates the body of the request, i.e., the

data, ensuring the validity of the data and providing a fast-fail logic to the clients. The data must be in JSON format with a predefined schema known to all involved components (i.e., IoT Gateway, Event Engine). The data are then transformed into a Kafka message and forwarded by the Kafka producer of the IoT Gateway to the "sensor–telemetry–event" topic. At this point, the data are valid, have a specific schema, and have been forwarded to the appropriate Kafka topic. Therefore, we are now talking about an IoT Event that carries measurements of the greenhouse accompanied by metadata helpful for their subsequent processing. The events, i.e., the messages forwarded and received to and from Kafka, may have a specific schema. However, the schema can support polymorphic data, thus allowing events destined for a purpose to be differentiated. In a complex and interconnected ecosystem, flexibility must be provided so that the requirements of one another do not bind each component. However, simultaneously, the validity and correctness of the data exchanged are ensured. Furthermore, the IoT Gateway does not necessarily know what constitutes correct information. It can verify the data, but their correctness is checked and managed by the Event Engine. The IoT Gateway acts as middleware to establish two-way communication with the IoT devices in the physical environment, i.e., the greenhouse and its peripheral structures. Finally, at this point, the raw data of the greenhouse are in the topic "sensor–telemetry–event" as IoT Event and can be consumed by the various components, most notably the Event Engine.

Up to this point, implementation is common and invariant for each presented scenario. It describes how conditions of the physical world, in our case air temperature, are measured, transformed, and sent from IoT Physical to Kafka to be made available by all components of the application architecture as events with the least possible semantics, i.e., as IoT Events.

Next, we detail our contribution, which is parametric Complex Event Processing through a simple microservice. More specifically, the microservice implements the Event Engine, which has parametric Complex Event Processing at its very core. Multiple microservices, as a whole, could also constitute the Event Engine. Technically, the Complex Event Processing implementations are based on the Kafka Streams "client" library [44], which offers a standardized and structured way to create stream processing applications. In addition, through the Object-Oriented paradigm offered by Java, CEP can be extended in terms of the functionality it offers following the standards delivered by the Kafka Streams library and without sacrificing the advantages of Kafka as a technology.

Event Engine's communication with Kafka is bidirectional and continuous. We could ping-pong where the Event Engine receives and sends messages continuously until it completes predetermined processing. Event Engine's implementation may consist of one or more event streaming applications. This approach aims to isolate the complexity at the implementation level and emphasize what input or inputs we want and what output or outputs we wish to receive. Conceptually, we apprehend it as a virtual synthetic functional interface with a detailed definition. This enables the various components and the user to produce and consume meaningful events, directly or indirectly, ignoring how they are created. Furthermore, each event streaming application can be parametric. Different instantiations can arise simultaneously and in parallel depending on the parameters. In the following sections, some examples are presented.

Next, we present and analyze the initial design and implementation of the Event Engine. We follow a bottom-up approach to create three abstractions for three well-known cases based on common sense and originating from the physical environment. It is noted that two of them are based on the scenarios presented in Section 6.

To form these scenarios in a bottom-up fashion, we use a template for defining a contract for creating an abstraction and its corresponding reusable implementation from the end user's point of view:

1. Which and what kinds of IoT devices are available in the physical environment?
2. Which and what kinds of functions can I create based on the available IoT devices?
3. What is the result I want to receive that will serve me in managing the physical environment?

Additionally, we use an elementary taxonomy to identify and define the factors that influence how a quantity (e.g., temperature) is ultimately converted into a meaningful event. Consequently, the very same factors that influence the formation of a meaningful event are the used by the user to parameterize the process of transformations. The factors that are related to IoT Physical include the number of sensors (one, many, all), the locations of sensors (which can be fixed or dynamic), and the business logic, which includes the design and management of the IoT Physical environment. Furthermore, the factors that are related to the Transformation Logic include the kind of computation (stateful, stateless), the calculation (e.g., statistic function, machine learning, scripting), the processing guarantees (e.g., at most once, at least once, exactly once), the time notion (e.g., event time, ingestion time, processing time), and the windowing (e.g., tumbling, hopping, sliding, session).

The template and the elementary taxonomy help create the Composite Transformations that we present next. Composite Transformations are designed and implemented from the building blocks of the Complex Event Processing model presented in Section 4. They are essentially reusable and independent event streaming topologies consisting of one or more event stream processors. Next, we present three Composite Transformations that serve needs that we will almost always encounter in a smart farming environment [45,46].

### 5.1. Composite Transformation: Ingestion

The motivation for creating the Ingestion Composite Transformation is ingesting measurements of natural quantities from sensors in the natural environment. The goal is to clean, validate, enrich, and homogenize the events. The user defines a vocabulary of physical quantities as parameters with individual details for each physical quantity. In this case, the details include instructions for converting measurements to a specific unit of measure, such as Fahrenheit to Celsius. As an extra degree of parameterization, the user also defines how physical quantities not included in the predefined quantity vocabulary are treated, for example, keep the events in a topic or discard them.

Each IoT device with sensing capabilities measures one or more things from the physical environment and sends them to the IoT Gateway. The IoT Gateway converts these data into an IoT Event with a specific schema that includes all measurements and valuable information, such as the IoT device ID and the measurement time. Furthermore, it promotes all instances of this type of IoT Event to a topic creating a single event stream. Therefore, a necessary abstraction is ingestion for sourcing and manipulating these coarse-grained primitive IoT Events into fine-grained IoT Events that each includes only one measurement and carries the minimum possible semantics. Moreover, transformed IoT Events are promoted to a different but defined topic, creating a single event stream. This enables further processing by other components without repeating these common transformations.

In the left side of Figure 3, an event stream is depicted that has records of the type "SensorTelemetryEvent" (i.e., all records have the same schema). The topic of this event stream is "sensor_telemetry_event". In the right side of Figure 3, the desired generated event stream is shown, with records of type "SensorTelemetryMeasurementEvent" (i.e., all records have the same schema). The topic of the generated event stream is "sensor_telemetry_measurement_event". That is, the second generated event stream includes a record for every measurement from every record of the first event stream.

This abstraction is parametric. The parameter is a list of the physical quantities to include in the generated event stream. The physical quantities that are not included in the list are discarded.

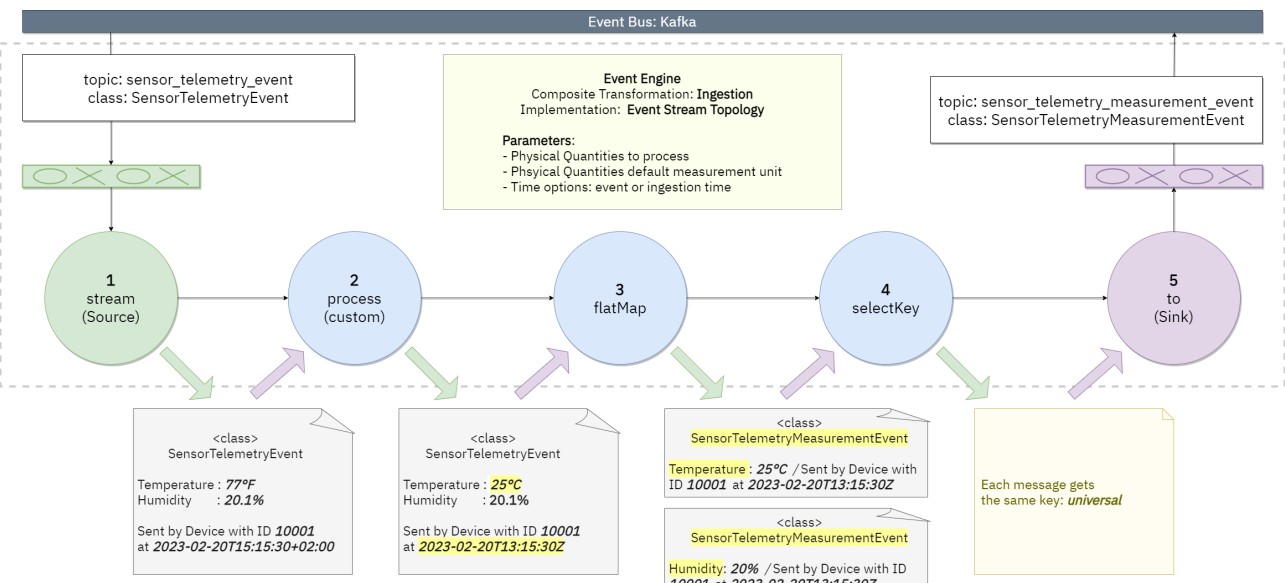

**Figure 3.** Event processor topology of Ingestion Composite Transformation.

In detail, the processing includes the following stages:

**1. stream**: Consumption of data stream from the topic "sensor_telemetry_event", i.e., the IoT Events, from the Event Bus sorted by the store timestamp. Also, this step includes deserializing the message payload from bytes in the format required by Event Engine, i.e., "SensorTelemetryEvent", which can be physically represented as either POJO or JSON. Comment: To determine the timestamp of the Kafka message, we use the custom implementation "TimestampExtractor" to ensure that each record has a correct and valid timestamp that will enable specific functions such as ordered consumption, dealing with late events, and windowing.

**2. process**: Pre-processing that involves converting specific values into the format required. More specifically, if an IoT Event does not have a timestamp, the processing timestamp is used. Second, since the IoT Event has a timestamp, it is checked for validity and then converted to a numeric representation as an epoch/Unix timestamp. That covers two prominent cases. First, the case where the IoT device does not include the event timestamp. Second, the case where the event timestamp is in a different format or in a different time zone than the one Event Engine requires. Event Engine only supports Unix timestamp as a universal convention so that there is homogeneity in the representation of instantaneous points in the timeline.

**3. flatMapValues**: Each "SensorTelemetryEvent" can contain more than one measurement. At this stage, new events of the type "SensorTelemetryMeasurementEvent" are created for each measurement included in the event, each of which now carries only one measurement. The metadata, as well as other information, such as the IoT device, the zone/location it belongs to, and the timestamp, are inherited from the original event. If a "SensorTelemetryEvent" has no measurement, it is considered valid and does not generate a new "SensorTelemetryMeasurementEvent", i.e., it does not have an effect.

**4. selectKey**: This specific stage concerns the assignment of a key (Kafka Message Key) for each event produced by the previous stage. The key selection concerns a technical aspect of the Event Bus, partitioning. We keep the example simple and set the same key for all generated events. At the same time, this option ensures that all events will be consumed in the order they were generated.

It is noted that the partitions and the choice of the message key are important factors that enable and affect scaling and concurrency. The example is simple and allows us to make basic settings. It is possible that some processing is conducted in one thread, thus reducing the throughput. Therefore, this option will work better in cases where a small workload is expected. Nevertheless, this option creates a benefit in terms of processing

order since Kafka guarantees that all events in a specific partition of a topic are consumed in order.

**5. to**: Each generated event is pushed to the "sensor_telemetry_measurement_event" topic. The result of the final stage is a new event stream with records of the type "SensorTelemetryMeasurementEvent".

It is noted that some processing nodes are activated for one or more records. Typically, messages are consumed by Kafka in batches with a fixed number for each batch at a specific time interval. These are technical details of Kafka and are fully configurable.

*5.2. Abstraction and Reusable Implementation: Splitting*

The motivation for creating the Splitting Composite Transformation is the selection of event types derived from the Ingestion Composite Transformation. That is, their separation from a single stream into distinct event streams, where each stream includes a specific type of measurement, i.e., physical quantity measurement. The specific Composite Transformation serves the requirement of creating homogenized event streams based on the physical quantity type. The user defines, as parameters, the physical quantities for which a discrete event stream must be created. The name of the topic (but also the topic itself), which includes the records of the event stream, is created dynamically. More specifically, the records of all event streams are self-contained and know certain information, such as the quantity type and others, as mentioned in previous sections. Therefore, they can feed processors with useful information to perform various processes. Each message gets the same key.

An additional abstraction, possibly to extend the first abstraction without depending on it, concerns splitting the events from the single event stream created by the above process. For each discrete physical quantity found in the events, a corresponding event stream is created with the corresponding topic (if it does not already exist). The purpose is to create topics that each include records related to the measurement of the same quantity.

This part of the abstraction is independent of the first part mentioned above. It is proposed for two main reasons: First, the creation of ready-to-use topics by consumers who want measurements of a specific quantity, and, second, the support of near real-time monitoring of one of the measurements of a specific quantity.

This abstraction is parametric. The parameter is a list of the physical quantities to create distinct event streams for.

In detail, the processing is depicted in Figure 4 and includes the following stages:

**1. stream**: Consumption of data stream from the topic "sensor_telemetry_measurement_event", i.e., processed IoT Events, from the Event Bus in order of the time sequence they were stored. In addition, this specific step includes deserializing the message payload from bytes in the format required by Event Engine, i.e., "SensorTelemetryMeasurementEvent", which can be physically represented as either POJO or JSON.

**2. filter**: This stage filters the records based on the physical quantity in which the measurement they carry refers to. More specifically, the condition is true if the physical quantity is included in the list of physical quantities the user selects to create discrete flows. Conversely, it is discarded if the physical quantity is not in the user's list. It is clarified that if discarded, the event continues to exist in the original event stream.

**3. to**: Each of the previously consumed events is checked to determine if the Event Engine supports the measurements, and if they are supported, the event is forwarded to the corresponding topic keeping its format, i.e., "SensorTelemetryMeasurementEvent". This stage supports the dynamic determination of the topic name. The format of the name is as follows: "sensor_telemetry_[NAME]_measurement_event" where "[NAME]" is the dynamic portion of the topic name replaced by the name of the physical quantity it represents.

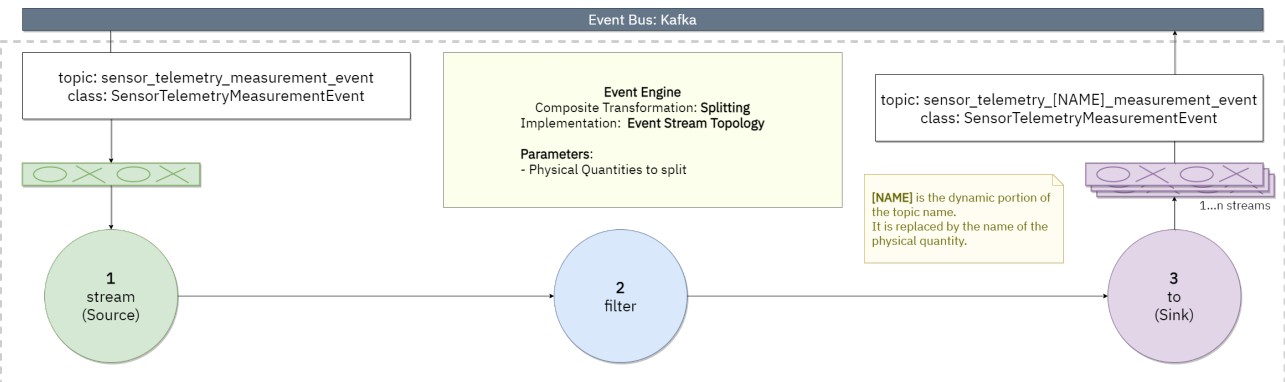

**Figure 4.** Event processor topology of Splitting Composite Transformation.

*5.3. Abstraction and Reusable Implementation: Time Windowed Moving Average Calculation*

Given a single homogeneous event stream with fine-grained events that represent different measurements of the same thing (or, rather, the same physical quantity), an abstraction is proposed to calculate the average of the last measurements of the specific physical quantity of each IoT device for a predetermined time window. Furthermore, the abstraction must take into account the minimum number of IoT devices required to calculate the average.

This abstraction is parametric. The parameters are the physical quantity, the duration of the time window, and the minimum number of IoT devices that must contribute to the calculation. Therefore, for each parameter set, a different instantiation of the specific abstraction results.

The result, commonly the desired output, is a meaningful Event for each time window with the average value, details about the time window, the contributing IoT devices, and more.

In detail, the processing is depicted in Figure 5 and includes the following stages:

**1 stream**: Consumption of data stream from the topic "sensor_telemetry_[NAME]_mea surement_event" where "[NAME]" is the dynamic portion of the topic name defined by the parameters and, more specifically, by the name of the physical quantity. The records of the specific event stream are of type "SensorTelemetryMeasurementEvent", and all concern the same physical quantity. It is noted that the event stream is homogeneous. It includes values in the same format, and all records are correct and valid. Any record that deviates is ignored from processing and managed via Dead Letter Queue. In addition, this specific step includes deserializing the message payload from bytes in the format required by Event Engine, i.e., "SensorTelemetryMeasurementEvent", which can be physically represented as either POJO or JSON. To determine the timestamp of the Kafka message, we use the custom implementation "TimestampExtractor" to ensure that each record has a correct and valid timestamp that will enable specific functions such as ordered consumption, dealing with late events, and windowing.

**2. groupByKey**: The stage to follow groups the records by the existing key. Since all records have the same key, only one sub-group will be generated. This technique is also known as global aggregation. Since we have a single sub-stream by assigning the same key to all records, later processing nodes will be executed by a single thread. This is very convenient in our prototype implementation, but it could scale better. Thus, we recommend this only when small workloads are expected. Grouping is also a prerequisite for windowing and/or aggregating a stream and ensures that data are adequately partitioned ("keyed") for subsequent operations.

**3. windowedBy**: The next step controls how the single sub-stream will be grouped into time windows for windowed aggregation, which is a stateful operation. We use hopping time windows which are windowed based on time intervals. They model fixed-sized overlapping windows, defined by two properties: the windows' duration and their advance interval. Since hopping windows can overlap, an event record may belong to

more than one such window. According to Kafka, hopping windows are sometimes called "sliding windows" in other stream processing tools. Kafka Streams follows the terminology in the academic literature, where the semantics of sliding windows differ from those of the hopping windows.

**4. aggregate**: At this point, records have been grouped into a single sub-stream and then regrouped into multiple windows according to the hopping time window definition. So, the stage to follow is the stateful processing of each windowed event stream. Each time a window is created, a data structure is initialized to hold calculations and other data throughout the time window. Each time a new event enters the windowed stream, it is checked whether it should be used for averaging. The statement that reflects the business logic is the following: If the value of the measurement has a timestamp that is greater than the timestamp of the measurement value of the last event for the given IoT device, then replace the measurement value in the stateful data structure and recalculate the average.

This stateful operation is a generalization of "reduce" and allows one to perform a complex stateful business logic operation and produce a single but different output for each input.

This processing produces a stream of key–value pairs, where the key contains information about the beginning and end of the time window. At the same time, the value, known as "AverageMeasurementValueAggregate", includes, among other things, the calculated average and all instances of "SensorTelemetryMeasurementEvent" that contributed to the calculation. Essentially, each record of this flow reflects a window with the corresponding data. Hopping time windows are aligned to the epoch (Unix timestamp), with the lower interval bound being inclusive and the upper bound being exclusive.

**5. suppress**: The abstraction prescribes the deletion of every intermediate change once the grace period is over. That is, only the final result for each time window will be emitted. Comment (limitation): Even after suppress operator is applied, the next event is required to advance the stream time to occur the final result.

**6. mapValues**: After a final result is emitted for a time window, it will be transformed into a Meaningful Event. More specifically, the new event includes the start and the end timestamps of the time window, the microservice identifier, the name and the unit of the physical quantity, the calculated average value, the instances of the "SensorTelemetryMeasurementEvent" that contributed to the calculation and last, but not least, information about the parameters of the instantiation of the microservice. Technically, the "AverageMeasurementValueAggregate" will be mapped to "WindowedAverageMeasurementValueEvent", and then the "WindowedAverageMeasurementValueEvent" will be enriched with the available information.

**7. selectKey**: See the source code and its comments for more information.

**8. to**: Each of the generated events, now Meaningful Events, are pushed to "windowed_average_[NAME]_measurement_value_event_[MICRO_SERVICE_ID]" topic, where "[NAME]" and "[MICRO_SERVICE_ID]" are the dynamic portions of the topic name and are replaced by the name of the physical quantity and the identifier of the microservice, respectively. The name of the physical quantity is included in the parameter set. At the same time, the microservice ID is automatically generated from the combination of the literal values of the parameters, i.e., a composite key. The result of the final stage is a new event stream with records of type "WindowedAverageMeasurementValueEvent".

To deal with the relative complexity induced by the dynamic creation of topic names for each instantiation of the proposed abstraction, we propose the gradual merging of event streams of the corresponding topics as follows:

1.  each physical quantity, creation of a topic "windowed_average_[NAME]_measurement_value_event" where "[NAME]" is the dynamic portion replaced by the name of the quantity. All the event streams generated by the microservices related to the specific quantity are merged in this topic.
2.  Creation of a topic "windowed_average_measurement_value_event" in which all the event streams generated from the previous step are merged.

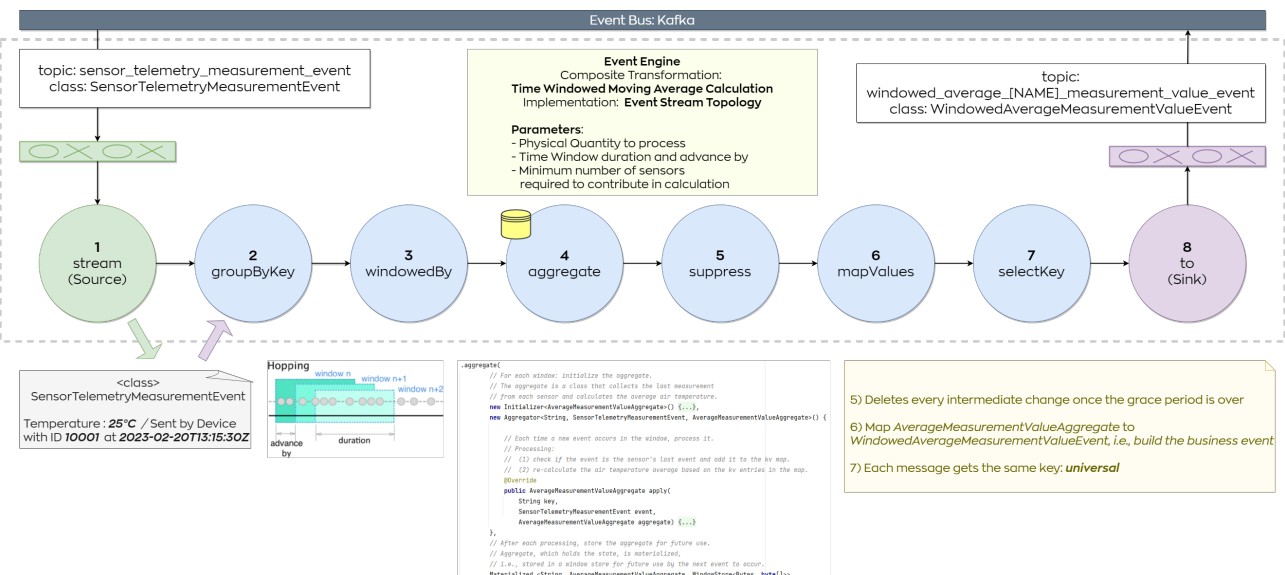

**Figure 5.** Event processor topology of Time Windowed Moving Average Calculation Composite Transformation.

These three abstraction implementations can be instantiated simultaneously and in parallel. Although there is a logical sequence between them, this does not limit the fact that they can operate independently. Thus, the flexibility provided by the proposed implementation of the application is demonstrated. Communication through events is based on conventions, while their processing is based on contracts. This ensures the system's evolvability, the separation of concerns, the composability, and preservation of the granularity of the functionality of each microservice. At the same time, it is technology-agnostic, thus supporting the heterogeneity emerging from an IoT ecosystem. On a technical level, starting a microservice requires two things: awareness of the name of the topic and of the schema imposed by either the topic or the consumer. Therefore, each instantiation of the above abstraction implementations can be used by any component that knows the source topic and the corresponding event schema.

The implementation is simple, and we purposely make no prior assumptions (in terms of modeling), no prior optimizations, and add no redundant complexity. Instead, driven by the needs and requirements derived from the scenarios, we implement precisely what is required and progressively add complexity as new needs and requirements emerge. The aim is to explore the practical implications of this research in depth and to document the practical challenges and possible solutions in detail.

## 6. Showcasing the Implementation of Parametric Event Transformation in a Smart Farming Domain

The aim of this section is to demonstrate the implementation presented in Section 5 in the smart farming domain. In essence, we aspire to show how the proposed architecture and the CEP model implementation would serve the requirements of a realistic [42,47] IoT-enabled application domain such as a smart farming one.

### 6.1. An IoT-Enabled Greenhouse: A Brief Description

We assume that there is a greenhouse with a roof vent through which the air temperature is controlled. When the door is open, warm air escapes from the greenhouse, reducing the temperature of the air in the greenhouse. When the door is closed, the temperature is maintained and gradually increases according to the weather conditions.

Farmers wish to control greenhouse conditions, in this case, air temperature, by opening or closing the roof vent for a specific or extended period of time. Their decisions are based on their knowledge, experience, and intuition. The goal of this setup is decision-

making based on information from the primary data produced by the sensors. Furthermore, automation of the opening or closing of the roof vent is an additional goal that is currently outside the scope of this research.

To make good and correct decisions, the farmers introduce smart management of the greenhouse by installing six sensors that, among other things, measure the air temperature and an actuator to open and close the roof vent automatically, mechanically, and remotely.

The data produced by the sensors may be helpful, but more are needed to make a decision. This is because the produced values are raw data that have yet to be transformed into meaningful information with domain-specific semantics. The CEP Model addresses this issue by providing a methodology, along with the functionality, for transforming IoT Physical events, i.e., data from the physical world, into meaningful business domain events, i.e., information and insights with semantics.

Next, based on the exact setup as shown above, we present some scenarios based on which the user, i.e., the farmer, wants to obtain the greenhouse's temperature. Although for each scenario, the main objective is for the farmer to know the temperature of the greenhouse, the way to calculate the temperature differs. For example, the temperature may refer to the whole greenhouse or a part of it; it may have been calculated differently (e.g., average, last temperature, maximum or minimum temperature, and more), and with a different number from sensors, it can be for a specific time window, and more. The basis for the scenario descriptions are the concepts presented in Section 4.

It is clarified that each scenario corresponds to a user requirement. Each scenario demonstrates the transformation pipeline in which events generated by sensors installed in the greenhouse are ultimately transformed into meaningful events. Moreover, each transformation is parametric, thus allowing farmers to test different ways of calculating temperature.

### 6.2. Scenario Events (IoT and Business)

#### 6.2.1. First Scenario

The first scenario involves calculating the temperature of the entire greenhouse considering data from one sensor. The final result, i.e., the final value of the temperature, is given in almost real time by the last sensor that sent data. That is, as the six sensors send data at different but close times, a data stream is created whose content is ignored, and only the last record is taken into account, as shown in Figure 6.

The first scenario is totally served by combining the *Ingestion* Composite Transformation and the *Splitting* Composite Transformation as presented in Section 5.

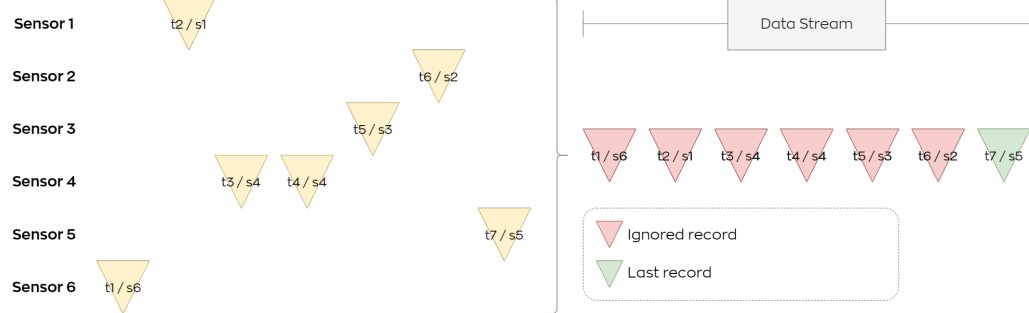

**Figure 6.** Event stream.

More specifically, this scenario is the instantiation of the parameterized *Ingestion* Composite Transformation and the *Splitting* Composite Transformation as presented in Section 5. The parameter is the physical quantity the farmer wants to monitor, i.e., the air temperature. The physical quantity parameter is applied to both Composite Transformations. The first Composite Transformation will perform the ingestion, and the second Composite Transformation will provide farmers with an event stream of monitoring events.

### 6.2.2. Second Scenario (with Three Variations)

The second scenario concerns the calculation of the greenhouse temperature as a result of the average of the last available temperatures of the corresponding sensors for a predetermined hopping time window as shown in Figure 7. In this case, we set the hop size to be one (1) minute and the window size to be five (5) minutes. The scenario has three variations that differ in terms of the number of temperatures of the respective sensors required to calculate the average and, finally, for the temperature produced to be valid.

In **variation A**, the final temperature is valid if at least one sensor has sent a temperature with an event timestamp within the predefined time window.

In **variation B**, the final temperature is valid if the majority of sensors, i.e., at least four sensors installed in the greenhouse, have sent a temperature with an event timestamp within the predefined time window.

In **variation C**, the final temperature is valid if at least six sensors, i.e., all the sensors installed in the greenhouse, have sent a temperature with an event timestamp within the predefined time window.

It is clarified that the variations are independent and do not overlap or negate each other. Although the description presents a common logical basis, the instantiations are independent and arise from the corresponding parameters, i.e., the physical quantity, the minimum number of sensors that must contribute to calculating the average, the time window hop size, and the time window duration.

The selection of the last temperature of each sensor for the predetermined time window is based on the event timestamp, i.e., the time when the sensor senses (measures) the data. More specifically, there may be none, one, or several temperatures in the predetermined time window for a sensor. Based on the logic of the scenario, only the last available temperature of each sensor (in the predefined time window) will be used if it exists (by contributing to the calculation of the average temperature).

It is noted that there is no triggering behavior in each window. The window is re-evaluated on the predefined fixed time intervals, independent of the actual content of the data stream (i.e., independent of the records). For clarity, a triggering behavior could include the re-evaluation of the temperature each time a new record occurs.

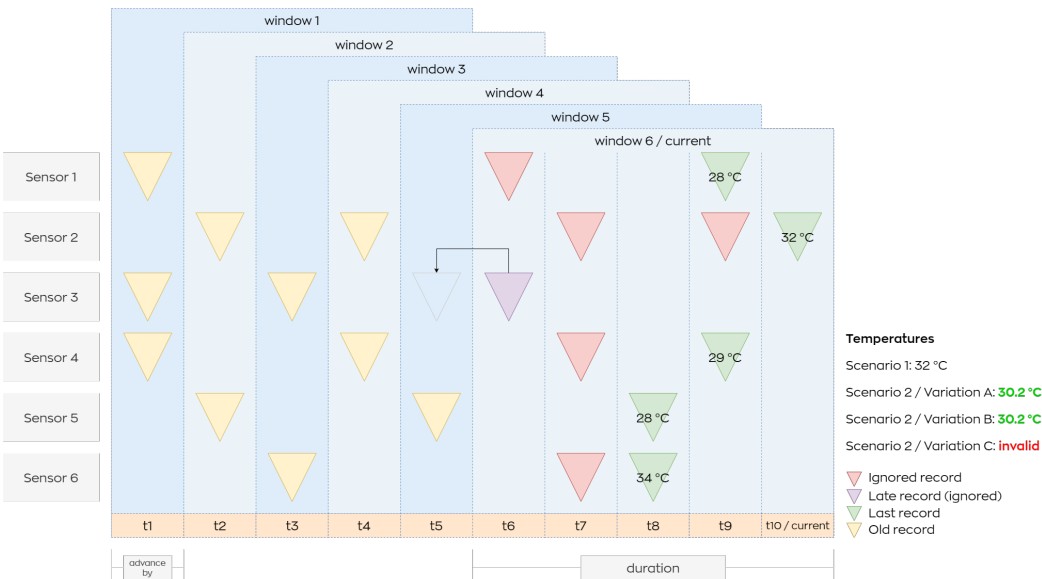

**Figure 7.** Hopping time windowed event stream.

The farmer can, of course, customize the execution of the scenario. In addition to variation selection, the standard user parameters for all variations are:

1. The hop size of the hopping time window (in seconds or minutes).
2. The window size of the hopping time window (in seconds or minutes).

3. The timestamp based on which the sorting will be done (event timestamp or ingestion timestamp).

For variations A and C, there are no other parameters. For variation B, the user parameter is the number of sensors taken into account for the calculation of the average (integer number from 1 to 6).

It is reminded that the calculation happens every minute. However, one more variation of this scenario would be for the computation mentioned above to occur every time a new IoT Event is generated, that is, a combination of the session time window and hopping time window, which would serve the requirement of knowing the temperature every time a sensor senses a change in the greenhouse, which represents a near real-time approach.

It is also clarified that a valid air temperature is a temperature measured, transmitted, and ingested in the prescribed time window.

The first scenario is totally served by the *Time Windowed Moving Average Calculation* Composite Transformation as presented in Section 5.

## 7. Discussion of Quality Properties

Provided in the literature are many quality properties to measure the Quality of Service in the IoT [48]; we focus on availability and reliability as two broad but vital concepts for commenting on the quality of a provided service. The emerging challenge from this discussion is to characterize the provided services in terms of how they serve the user's requirements under different conditions in the physical environment. This increases user engagement by contributing to the creation of a flexible user-centric architecture. Additionally, it is essential to comment on the aforementioned quality properties and set future directions for creating a QoS framework for the IoT. At the same time, this specific discussion validates the reason for the existence of the proposed architecture and the CEP model, significantly strengthening its guarantees for the adoption of these practices in a production environment.

According to [49,50], there are many quality properties, including accessibility, availability, elasticity, installability, interoperability, maintainability, privacy, reliability, resource utilization, responsiveness, scalability, security, testability, and usability. In this paper, we focus on two basic quality properties that can provide fundamental validation of the quality of a provided service based on the overall behavior of the system. These quality properties are Availability and Reliability [49,50]. Studying the quality properties and analyzing trade-offs helps users make decisions to flexibly and effectively manage IoT Physical under different circumstances and situations. Moreover, these particular quality properties correspond with the extent to which user needs are satisfied when the system is used in a particular operation and the extent to which the system can adapt to new operational conditions. Formally, this is the extent to which the system provides functions that meet stated and implied needs when used under specified conditions (bottom-up).

In this paper, a significant challenge emerges related to the quality properties each Composite Transformation may or may not serve. Each Composite Transformation serves specific quality properties that fit different use cases based on the requirements of the user and the conditions of the physical world. Likewise, each scenario serves specific quality properties to a different degree and is based on certain conditions, which are known and generally maintained for each execution of the scenario without it being binding, since an IoT ecosystem is naturally complex and often unpredictable.

When we study an IoT system, many factors [48–51] influence certain metrics, whether qualitative or quantitative. Moreover, these factors usually concern all layers of a typical IoT architecture [6,52]. However, in the example and scenarios we present in Section 6, we focus on two basic non-ideal conditions regarding IoT Physical. More specifically, the errors we study are (i) sensor failure that includes physical damage, disconnection, or any other reason that may set the sensor unavailable and (ii) indefinite delay of data transmission from a functional and available sensor. Our choice is made to preserve the simplicity of the example but also to record and highlight how we deal with the increasing complexity

that basic real-world situations may introduce. Next, we discuss the quality properties per scenarios we have already presented.

### 7.1. First Scenario

The first scenario offers a high degree of **Availability**, since there is no single point of failure in the greenhouse. In particular, if one sensor is unavailable, the others will continue to sense and produce temperature data and thus the temperature value. The condition in which the service is unavailable is when all the sensors installed in the greenhouse stop working.

On top of that, the scenario offers a fault-tolerant service since, in any case in which the condition mentioned earlier is not met, the service will continue its intended operation. Furthermore, it is noteworthy that the user knows the sensors that are not in operation, since they can see each sensor when it sends data for the last time. So the user can fix physical-world hardware faults when they arise.

The **Time Behavior** is served at a high degree. More specifically, the response time is low, and the turnaround time is also low since, in IoT, minimal and fundamental stateless transformations will occur. Therefore, the total time for generating the temperature value is short. In addition, given that the processing in the Event Engine is simple and fast and that the Event Bus of the architecture offers extremely high throughput, the scenario has high throughput to boot.

However, there is a non-ideal condition that, when satisfied, affects the timeliness of the generated temperature value. Specifically, the user makes decisions based on the ingestion timestamp instead of the event timestamp. As a result, the timestamp of the generated "meaningful" event will be the ingestion timestamp of the IoT Event. So when a sensor is late in sending data, the Event Engine produces a "meaningful" event with a correct and valid temperature value based on the concepts defined by reliability. Nevertheless, this value differs from the actual one because time has passed. It is recalled that, in any case, the user is aware of this information and can either take additional actions or wait for the next actual temperature value, which, based on this scenario, will be very soon.

Moreover, this scenario offers a relatively high degree of **Time Behavior**: It is inclusive, since it takes into account all the sensors, even though they do not collectively contribute to the calculation of the temperature value. The validity and correctness of the input data always result in the production of a valid and correct temperature value. Given that the condition mentioned previously in the availability analysis is maintained, i.e., at least one sensor is available, the incoming events from the physical environment are always sufficient to calculate the temperature value.

At the same time, the functional correctness is served to the absolute degree, since when at least one sensor is available (i.e., there is not a total lack of sensors), business continuity of the service is guaranteed. Nevertheless, it is evident that when one of the two non-ideal conditions arises, the "meaningful" events resulting from the service may not be trustworthy for the user.

### 7.2. Second Scenario

Here, the evaluation of each variation based on the quality properties is presented.

First, all three variations have some common reference points concerning the aspects of quality properties we are considering.

The **Time Behavior** is simply the same for all three variations. More specifically, the time required to calculate the average temperature is clearly increased, since several gradual and sequential stateless and stateful processing steps are required. Therefore, although the response time remains constant, the turnaround time is relatively increased. Furthermore, the total time required for the final temperature is affected to the greatest extent by the turnaround time; therefore, it will also be increased.

Regarding the timeliness of the service, it is affected by two main issues. First, the total processing time, which, as mentioned, will be relatively increased. Second, the temperature

calculation is conducted every minute. Therefore, any change in the sensor measurements will not be immediately visible in one-minute intervals. Instead, it will be visible shortly after the end of the one-minute interval. However, the final result, when obtained, primarily reflects greenhouse conditions, since the event timestamp is taken into account and not the ingestion timestamp. If a sensor sends delayed data while it had previously—but always within the time window—sent data at the correct time, the delayed data will not be considered in calculating the average.

Regarding fault tolerance as an aspect of **Availability**, each variation has a different tolerance for IoT Physical errors. However, one factor of fault tolerance related to late but valid events within the predefined time window is the same for all three variations. So, the additional, common aspect contributing to fault tolerance is that the event timestamp is taken into account instead of the ingestion timestamp, which means that late events will be ignored or not contribute to the average calculation. Given that there is at least one correct temperature in the predefined time window for each sensor in the total number of sensors required for each variation, the service is available.

Regarding trustworthiness as an aspect of **Reliability**, given that the time window of one minute is small, the result of the service is trustworthy. However, even in the interval of one minute, there are significant changes, and the user is not aware of them in real-time, but in a short period of time, they receive knowledge of the final temperature as well as of all the temperatures that contributed to its calculation.

The most important, conserved, common reference point is that in all three variations when the service is available, reliability is served to the absolute extent.

Some additional common references points include:

1. The service is inclusive, since it includes (or tries to include) all the available sensors.
2. The validity and correctness of the input data always result in the production of a valid and correct temperature value.
3. Late events, i.e., invalid events, do not affect the validity and the correctness of the calculated temperature, since IoT Events are sorted based on the actual timestamp, i.e., the event timestamp.
4. To the extent that the service is available, functional correctness is maintained in both cases of the examined faults.

### 7.2.1. Variation A

Service availability based on scenario variation is maintained when at least one sensor sends a valid air temperature in the prescribed time window. The service is unavailable if all sensors do not measure or send a valid air temperature in the prescribed time window.

The service is fault-tolerant as long as at least one sensor is working. In the case of a non-total error, the service is available by calculating the average with the available sensors. The same applies to functional correctness, as the service is functional in any case that does not include a non-total error.

### 7.2.2. Variation B

Service availability based on scenario variation is maintained when the majority of the sensors (four sensors) send a valid air temperature in the prescribed time window. The service is unavailable if three or more sensors do not measure or send a valid air temperature in the prescribed time window.

The service is fault-tolerant as long as the majority of the sensors are working. In case of an error that does not include three or more non-working sensors, the service is available by calculating the average with the available sensors (four, five, or six). The same applies to functional correctness, as the service is functional in any case that does not include an error in three or more sensors.

### 7.2.3. Variation C

Service availability based on scenario variation is maintained when all sensors send a valid air temperature in the prescribed time window. The service is unavailable if at least one sensor does not measure or send the valid air temperature at least once in the prescribed time window.

The service is not fault-tolerant, since the final temperature cannot be calculated in case of an error of at least one sensor. The functional correctness is maintained only in one of the two cases of the examined errors, i.e., when a sensor is late in sending a valid air temperature.

## 8. Conclusions

In summary, we present a symmetrical architecture that enables two-way communication between IoT devices and Applications and Clients and, thus, allows users and systems to understand and manage a physical environment. Furthermore, we extend the possibilities of the architecture with a CEP model for transforming IoT events into meaningful business events. Based on the proposed architecture and the CEP model, we provide an implementation that could serve any IoT-enabled application context. We exemplify it with a smart farming case in an IoT-enabled greenhouse. The implementation includes a library of Composite Transformations that each describes, in detail, the gradual and sequential steps for transforming an event. Specifically, three fundamental Composite Transformations are included: ingestion, event splitting, and calculation of the average value of measurements. The appropriateness and possibility of inclusion and integration of the implementation in a production environment are demonstrated by showcasing the functionality in a smart farming domain through four scenarios that each reflect a user's requirements. In addition, we comment on and discuss the quality properties trade-offs that increase user engagement by allowing them to adapt the provided services based on their requirements, which arise mainly from the conditions that prevail in the natural environment.

Ultimately, this paper contributes with implementation based on the proposed architecture and CEP model that follows a highly flexible bottom-up approach (since all data originate from IoT or Application layers) to create IoT services. These IoT services collectively provide the highest possible visibility and control over a complex system, thus increasing user engagement. As a result, this approach may significantly impact the overall architecture and its evolution. In fact, it is intended that our architecture interact with a large number of distributed and heterogeneous event sources and sinks (e.g., sensors and actuators, respectively) that observe and operate in any application context. This is typical of most CEP scenarios, such as environmental monitoring, business process automation, and control systems. From another point of view, our CEP model and its implementation show how we eventually transform the IoT events to business events, reflecting the requirements and knowledge of the IoT-enabled business domain. The proposed implementation could be duplicated, expanded, and deployed in IoT domains with similar business requirements by researchers and practitioners interested in IoT architectures and systems. For example, this implementation could easily, and without modification, serve in temperature monitoring in a smart building, such as a house or even a hospital.

Our architecture can already manage and interact with IoT devices in any IoT-enabled domain, such as smart city, smart industry; however, we still need to realize some limitations that we intend to address in the future. We aspire to make the data distribution in the architecture more scalable, since this is a main requirement for IoT architectures. Moreover, we want to extend the CEP model to support common CEP patterns and also extend the architecture to support actuation in the IoT physical context, i.e., implementation of the process engine. We intend to make such improvements to ensure the proposed architecture's generalizability. Last but not least, we could also study the quality properties of the architecture.

**Author Contributions:** All authors contributed equally to this work. All authors have read and agreed to the published version of the manuscript.

**Funding:** The research presented has received funding from the European Union's Project H2020 PHYSICS (GA 101017047).

**Data Availability Statement:** Not applicable.

**Conflicts of Interest:** The authors declare no conflict of interest.

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
