# Peer review of "Transforming IoT Events to Meaningful Business Events on the Edge: Implementation for Smart Farming Application"

_futureinternet, doi:10.3390/fi15040135_

Round 1
Reviewer 1 Report
This paper proposed a novel symmetric architecture that enables two-way bidirectional communication in IoT and integrates Complex Event Processing to transform IoT events into meaningful business events. It also shows the details of the implementation of Smart Farming focusing on an IoT-enabled greenhouse. This paper is well written, and the details of the idea are well described. Some improvements on the structure are needed, e.g., In Section 7, it is recommended to use sub-sections for different scenarios.
Reviewer 2 Report
This is a nice paper with very detailed descriptions of the architecture and the approach of CEP. Apart from a few typos etc., I suggest answering/addressing the following questions/suggestions:
- It does not fully become clear if the architecture has been presented before in reference [43] and if the text in Chapter 3 is only a summary. Please make this clear at the beginning of the chapter.
- In the conclusions, you say that it is not clear whether your architecture can be used in other IoT domains. Why do you think this is the case? The architecture seems very general to me, and I wonder what would prohibit a use in other domains.
Apart from that, I believe the paper can be published more or less as is.
